# Predictive modeling of lean body mass, appendicular lean mass, and appendicular skeletal muscle mass using machine learning techniques: A comprehensive analysis utilizing NHANES data and the Look AHEAD study

**Daniel Olshvang**[1]*, **Carl Harris**[1], **Rama Chellappa**[1,2], **Prasanna Santhanam**[3]

1 Department of Biomedical Engineering, Johns Hopkins University, Baltimore, MD, United States of America, 2 Department of Electrical and Computer Engineering, Johns Hopkins University, Baltimore, MD, United States of America, 3 Division of Endocrinology, Diabetes, and Metabolism, Department of Medicine, Johns Hopkins University School of Medicine, Baltimore, MD, United States of America

* dolshva1@jhu.edu

## Abstract

This study addresses the pressing need for improved methods to predict lean mass in adults, and in particular lean body mass (LBM), appendicular lean mass (ALM), and appendicular skeletal muscle mass (ASMM) for the early detection and management of sarcopenia, a condition characterized by muscle loss and dysfunction. Sarcopenia presents significant health risks, especially in populations with chronic diseases like cancer and the elderly. Current assessment methods, primarily relying on Dual-energy X-ray absorptiometry (DXA) scans, lack widespread applicability, hindering timely intervention. Leveraging machine learning techniques, this research aimed to develop and validate predictive models using data from the National Health and Nutrition Examination Survey (NHANES) and the Action for Health in Diabetes (Look AHEAD) study. The models were trained on anthropometric data, demographic factors, and DXA-derived metrics to accurately estimate LBM, ALM, and ASMM normalized to weight. Results demonstrated consistent performance across various machine learning algorithms, with LassoNet, a non-linear extension of the popular LASSO method, exhibiting superior predictive accuracy. Notably, the integration of bone mineral density measurements into the models had minimal impact on predictive accuracy, suggesting potential alternatives to DXA scans for lean mass assessment in the general population. Despite the robustness of the models, limitations include the absence of outcome measures and cohorts highly vulnerable to muscle mass loss. Nonetheless, these findings hold promise for revolutionizing lean mass assessment paradigms, offering implications for chronic disease management and personalized health interventions. Future research endeavors should focus on validating these models in diverse populations and addressing clinical complexities to enhance prediction accuracy and clinical utility in managing sarcopenia.

**Data Availability Statement:** The NHANES data utilized in this study is publicly available. The collated input data for the relevant section of the article can be accessed via DOI: https://doi.org/10.6084/m9.figshare.25215284. The data from the Look AHEAD: Action for Health in Diabetes study [(V9)/https://doi.org/10.58020/wr3g-1218] reported in this article are available upon request from the NIDDK Central Repository (NIDDK-CR) website, Resources for Research (R4R), at https://repository.niddk.nih.gov/.

**Funding:** The author(s) received no specific funding for this work.

**Competing interests:** The authors have declared that no competing interests exist.

## Introduction

Sarcopenia is a skeletal muscle disorder characterized by loss of muscle mass, skeletal muscle function, strength, and quality, often measured in relation to gait speed [1, 2]. It is associated with a myriad of negative health outcomes including increase morbidity, mortality, and overall low quality of life, irrespective of the definitions used, independent of population studied, especially in persons with cancer and in the geriatric population [3, 4]. In addition, sarcopenia is a predictor of morbidity and mortality following abdominal surgery and cardiovascular diseases [2, 5]. The reduction in muscle quantity and function either compounds or results from problems of weight loss, reduced food intake, low BMI (Body Mass Index), anorexia, poor physical performance often associated with conditions contributing to malnutrition like advanced malignancy and chronic diseases like chronic obstructive pulmonary disease (COPD), chronic kidney disease and heart failure (termed cachexia) [1]. Sarcopenia remains unrecognized and poorly evaluated, largely because of lack of large-scale data and universal applicability [6]. Predicted lean body mass (LBM) is thought to explain the 'obesity paradox'—the intriguing phenomenon of lower mortality in individuals with higher BMI, often depicted by a J-shaped curve, defying conventional beliefs about the negative impact of obesity on health outcomes [7].

Dual-energy X-ray absorptiometry (DXA) is a simple and invaluable tool for estimation of axial and appendicular lean mass, fat free mass, Bone Mineral Content/Bone Mineral Density (BMC/BMD) and muscle mass (inferentially) [8–11]. Despite this evidence, routine DXA is limited to estimating bone mineral density, and bone mineral content at specific sites (Lumbar spine, forearm, and hip) as recommended by osteoporosis guidelines [12]. Of particular interest and clinical importance is the appendicular lean mass (ALM), which is the sum of lean tissue in the arms and legs, often reported alone or adjusted to height or BMI [10]. Appendicular skeletal muscle mass (ASMM) and ALM are associated with gait speed, strength -metrics that are important components of sarcopenia.

Unique equations have been designed and validated for estimation of LBM, ALM and ASMM from bioelectrical impedance analysis (BIA) measurements, with DXA as the reference standard [13]. Machine learning algorithms tried and evaluated include Light GM, random forest, XGBoost, K-nearest neighbors (KNN) and support vector machines (SVM) [14]. However, the entire cohort consisted of around 160 individuals aged over 65 years in a community setting, which is clearly not representative of the general population. Machine learning has also been used to predict sarcopenia from electronic health records [15]. However, the study was a specialized tertiary center experience, involved musculoskeletal testing -grip strength, chair stand strength, short physical performance and then finally ASMM measured by DXA.

In this study, we address these gaps by utilizing a large and diverse dataset from the National Health and Nutrition Examination Survey (NHANES) and applying the findings to the Look AHEAD study—a substantial prospective endeavor funded by the National Institutes of Health (NIH). The aim is to predict LBM, ALM, and ASMM normalized to weight using a combination of simple anthropometry and DXA-derived metrics. A comparative analysis, encompassing Machine Learning (ML) techniques and established anthropomorphic equations [16], aims to discern the most effective models. This comprehensive approach facilitates a nuanced understanding of predictive capabilities and their potential application in diverse populations, moving beyond the confines of specialized settings.

## Materials and methods

This study utilized a secondary analysis of data from two large-scale studies: the National Health and Nutrition Examination Survey (NHANES) and the Action for Health in Diabetes

(Look AHEAD) study. The NHANES data, spanning from 1999 to 2018, served as our primary dataset for model development and initial testing. The Look AHEAD study data was used for external validation.

## Study population and data sources

The inclusion criteria for our cohort involved subjects between 45–75 years who had undergone a DXA scan for body composition. Pregnant women were excluded from the cohort. We used the terms "height," "weight," and "waist circumference" in this analysis for clinical clarity. However, we acknowledge that the International Society for the Advancement of Kinanthropometry (ISAK) recommends the use of "stature" for height, "body mass" for weight, and "waist girth" for waist circumference to align with the technical definitions in anthropometry.

### NHANES

The National Health and Nutrition Examination Survey (NHANES) provides comprehensive, cross-sectional data encompassing various demographic groups, including key metrics like weight and waist circumference. Researchers have frequently utilized this dataset in their studies over the years [17–19]. Our analysis focused on the NHANES CDC (Centers for Disease Control) dataset, spanning two decades from 1999 to 2018. NHANES data, available publicly and deidentified, serves as a crucial resource for numerous epidemiological studies. It has been instrumental in establishing standard reference ranges for diverse laboratory tests, physical examinations, and imaging techniques.

The DXA protocol utilized a Hologic (QDR-4500A) fan beam densitometer for whole-body scans. Certified radiology technologists conducted these examinations at mobile examination centers, following detailed protocols from the NHANES Body Composition Procedures Manual. Robust quality assurance involved meticulous phantom scanning schedules, technologist performance monitoring, and maintenance by Hologic engineers. University of California San Francisco analyzed scans using Hologic Discovery software, employing expert review and invalidity codes to ensure accuracy. Quality control scans and cross-calibration studies verified system uniformity and densitometer performance consistency across multiple sites, crucial for data pooling in NHANES. Longitudinal monitoring detected and corrected any scanner-related changes, while a comprehensive quality control program addressed issues promptly, enhancing overall scan accuracy and consistency [8].

### Look AHEAD

The Action for Health in Diabetes (Look AHEAD) study, an extensive and prolonged clinical trial overseen by the National Institute of Health (NIH) in the United States, was designed as a randomized controlled trial. Its primary aim was to investigate how intensive lifestyle interventions could impact the health and overall well-being of individuals diagnosed with type-2 diabetes [20, 21]. The Look AHEAD dataset specifically focuses on overweight individuals diagnosed with type-2 diabetes, encompassing both initial baseline information and subsequent follow-up data collected over an extensive 8-year period. However, for this analysis, only the baseline data was used, employing identical variables as those used in the NHANES dataset. By working with a dataset that incorporates a domain shift due to different study population, we can evaluate our models' generalizability.

DXA measurements for Body Composition were conducted at four Look AHEAD sites, utilizing Hologic (QDR-4500A) fan beam densitometers. Software updates were endorsed by a DXA quality assurance center (University of California San Francisco) during the study period. Cross-calibration phantoms circulated at baseline to assess scanner disparities, while

ongoing monitoring with spine and whole-body phantoms ensured longitudinal consistency. Corrections based on whole body phantom data were applied to participant body composition findings. Additionally, Hologic software adjusted whole body scan outcomes to account for potential underestimations in fat mass. Central monitoring guaranteed acquisition and analysis quality, with participants exceeding three hundred pounds excluded due to DXA scanner weight limitations. These assessments adhered strictly to the guidelines outlined in the DXA Quality Assurance Operations Manual [22].

## Ethical considerations

As our analysis involved utilizing pre-existing data from both NHANES and Look AHEAD studies, it was deemed eligible for exemption from Institutional Review Board (IRB) review.

## Dataset comparison

Comparisons between NHANES and Look AHEAD datasets were conducted to analyze variations in body composition metrics. Since the Look AHEAD dataset includes only individuals with type-2 diabetes, while the NHANES dataset includes the general population, it inherently incorporates a domain shift. Variations in model performance and any disparities in body composition between these datasets were meticulously examined.

## Evaluation of body composition

In a puritanical world, fat-free mass (FFM) implies all body components except fat, such as bones, muscles, hydration (water content) and organs. It is measured using techniques like dual-energy X-ray absorptiometry (DXA) or estimated with bioelectrical impedance analysis by using equations that incorporate variables such as impedance, resistance, and reactance [23]. FFM is different from LBM as a small fraction of total body weight (up to 3% in men and 5% in women). lipids in cellular membranes are included in LBM [24]. According to Heymsfield et al, LBM excludes bone mass, focusing instead on muscles and organ tissues [25]. However in published literature, LBM may include Bone Mineral Content (BMC) or exclude it [26, 27].

Skeletal muscle mass (SMM) specifically targets the contractile tissues and is typically estimated from lean soft tissue measurements using predictive equations from MRI [28]. However, SMM derived from equations involves the use of MRI and one of the two methods. One method involves setting a threshold for adipose (fat) and lean tissue based on the gray-level histograms of the images. The other method uses a filter to differentiate between various gray-level regions in the images, and then applies a watershed algorithm to outline these regions [29]. However, such techniques are impractical in large population cohorts.

Technically, LBM and FFM are distinct terms, as LBM includes FFM plus essential fat in the tissue, which varies between 2% and 10% of FFM. LBM measured by DXA is actually closer to FFM (with or without bone minerals), making it quantitatively less than FFM [29]. Despite these shortcomings, DXA is considered the gold standard for measurement of muscle mass [30]. We have used datasets (NHANES/Look AHEAD) that have used DXA based methods for body composition analysis. In addition, we also performed computations to adjust for the presence of FFAT (fat free adipose tissue), as outlined in this seminal paper by Takashe Abe et al. [31]. Presence or absence of sarcopenia is determined by the extent of FFAT [32].

First, we calculated adipose tissue mass using DXA-derived fat mass because 85% of adipose tissue is fat (i.e., adipose tissue = fat mass ÷ 0.85). Then, FFAT was calculated as FFAT = adipose tissue × 0.15. Finally, we computed adjusted total lean body mass (TLM),

adjusted appendicular lean mass (ALM) which includes BMC, and adjusted appendicular skeletal muscle mass (ASMM) which excludes BMC, after subtracting FFAT from the respective compartments [31]. Since there is no organ tissue in the limbs, ASMM can be reliably estimated by subtracting BMC from ALM.

Our primary outcomes were TLM, ALM, and ASMM, all derived from DXA scans and normalized to body weight and adjusted to the presence of FFAT.

## Data preprocessing

The NHANES data spanning the years 1999–2018 was combined and organized into tabulated formats. Subsequently, the data underwent division into training and testing sets, employing an 80/20 split. Following this, standardization was applied, preceded by the utilization of one-hot encoding to generate dummy variables for categorical data.

The predictive variables for total lean mass were age, ethnicity, weight, height, and waist circumference. Due to slight differences in the encoding of race/ethnicity between NHANES and Look AHEAD, a consolidation was performed, creating dummy variables for Hispanic ethnicity, White/Black race, and other/mixed race.

In the NHANES dataset, the initial subject count was 45,411. Subjects with missing data for any of the specified features, as well as subjects without fat mass data in their DXA scan necessary for the adjusted mass calculation were excluded. Additionally, an age filter (45–75) was applied to align with the Look AHEAD age range. This resulted in a final sample size of 11,061. For the Look AHEAD dataset, the initial sample size was 4,906. However, only 1,369 subjects had DXA scan results, which served as the ground truth for the analysis, establishing the final sample size.

## Post-prediction processing

The anthropometric equations and all the models were used to predict the weight of the lean mass–TLM, ALM, and ASMM–in grams, as measured by the DXA scan. After the prediction task, all values were normalized by their corresponding body weights to provide mass percentages. This would allow the comparison of lean mass adjusted for body weight.

## Statistical analysis

All analyses were performed using Python 3.11.3 with the scikit-learn 1.2.2, XGBoost 2.0.0 and Lassonet 0.0.14 libraries. For each model, we calculated the Mean Absolute Percentage Error (MAPE) and coefficient of determination ($R^2$) as our primary evaluation metrics. We also computed the Root Mean Square Error (RMSE) for each model.

To assess differences between the training, testing, and validation datasets, we employed Yuen-Dixon tests for continuous variables and chi-square tests for categorical variables to compare the characteristics of our training/test set with our validation set. These tests were chosen to robustly assess differences in distributions between the two datasets.

## Evaluation metrics

**Mean Absolute Percentage Error (MAPE).**   MAPE is a crucial metric particularly suited for assessing the accuracy of models predicting percentages. It calculates the average percentage difference between predicted and actual values. This metric offers a comprehensive understanding of the average magnitude of percentage errors in the predictions [33]. Since we evaluate the predictions after they have been normalized to provide lean mass percentages, the

MAPE calculation can be described as the following:

$$MAPE = \frac{1}{N} \sum_{i=1}^{N} 100 * \frac{|y_{pred}^{(i)} - y_{true}^{(i)}|}{w^{(i)}} \tag{1}$$

Where $y_{pred}^{(i)}$ is the $i$th predicted value (in kg), $y_{true}^{(i)}$ is the $i$th actual value (in kg), $w^{(i)}$ denotes the total weight of the $i$th sample (in kg) and $N$ denotes the sample size. The average distance between the values provides the final MAPE value.

In our study, MAPE played a significant role in evaluating the accuracy of predictive models for estimating lean mass percentage. By focusing on percentage errors, MAPE provided insights into the relative accuracy of predictions, enabling a more intuitive assessment of model performance in this context.

**Coefficient of determination ($R^2$).** $R^2$, or the coefficient of determination, measures the proportion of variance in the dependent variable (lean mass percentage) explained by the independent variables within the model. It ranges from 0 to 1, with one indicating a perfect fit. $R^2$ is an essential metric to assess the model's ability to explain variability in the observed data [34], and is defined as:

$$R^2 = 1 - \frac{\sum_{i=1}^{N} \left(y_{true}^{(i)} - y_{pred}^{(i)}\right)^2}{\sum_{i=1}^{N} \left(y_{true}^{(i)} - y_{mean}^{(i)}\right)^2} \tag{2}$$

Where $y_{pred}^{(i)}$ is the $i$th predicted value (in kg), $y_{true}^{(i)}$ is the $i$th actual value (in kg), $y_{mean}$ is the population mean (in kg), and N denotes the sample size.

In our study, $R^2$ was pivotal in determining the goodness of fit of predictive models, indicating the strength of the relationship between predictors and lean mass percentage. A higher $R^2$ value signified a more robust model with increased explanatory power.

**Utilization of metrics in model assessment.** The incorporation of MAPE and $R^2$ facilitated a comprehensive evaluation of predictive models developed using NHANES data for estimating lean mass percentage. These metrics collectively provided a multifaceted assessment of model accuracy, precision, and explanatory power. While MAPE gives an insight on the accuracy of the prediction by targeting the distance between the predicted value and the actual value, $R^2$ provides an understanding on how much does the model account for the variance of the prediction tasks. Their use guided the identification of the most effective models, enabling the selection of robust models for further analysis and validation.

## Models and techniques

**Anthropometric equations.** Multiple studies have been conducted, focusing on prediction equations to estimate muscle mass using anthropometric data [35]. A recent and well-validated study using this method is the study by DH Lee et al. [16].

In this NHANES-based study, anthropometric equations were formulated and validated for estimating lean body mass. Adult participants underwent DXA measurements to assess body composition, collecting standardized anthropometric data such as height, weight, BMI, waist circumference, and additional measures. Prediction equation development involved multivariable linear regression, revealing that a basic model with sex, age, height, and weight and waist circumference explained most of lean body mass variation. Polynomial terms and interactions provided minimal benefit. Validation using a separate NHANES group demonstrated the equations' robust predictive accuracy for lean body mass in both sexes, comparable to DXA-measured values. By comparing with additional biomarkers such as serum creatinine levels, the prediction equations provide a more validated

approach than simple linear regression. However, the equations developed were not validated on separate datasets.

As a result, we implemented the following equations:

For males:

$$TLM = 19.363 + 0.001*age + 0.064*height + +0.756*weight - 0.366*waist_{circ} \\ + 0.231*eth_{hisp} + +0.432*eth_{black} - 1.007*eth_{other} \tag{3}$$

For females:

$$TLM = -10.683 - 0.039*age + 0.186*height + +0.383*weight - 0.043*waist_{circ} \\ - 0.059*eth_{hisp} + +1.085*eth_{black} - 0.34*eth_{other} \tag{4}$$

Where $eth_{hisp}$, $eth_{black}$, $eth_{other}$ are categorical variables representing Hispanic, Black or African-American, and Other/Mixed ethnicities, respectively.

**Linear regression.** This model serves as a fundamental tool in predictive analytics, aiming to establish a linear relationship between the input features and the target variable. It assumes a linear association between predictors and the predicted outcome, which in our case, is the lean body mass. We utilized this model as a baseline method due to its simplicity and interpretability. The model fitting involves minimizing the sum of squared differences between the predicted and actual values [36].

**Random forest.** This ensemble learning technique constructs numerous decision trees and combines their outputs to generate final predictions. Each tree in the forest is built independently, and the final prediction is derived from an aggregation of individual tree predictions. Random Forest is known for its' robustness to overfitting and ability to handle complex relationships in the data, making them suitable for predicting lean body mass in our study [37].

We implemented a Random Forest regression model with 100 trees with no limitation on tree depth and used mean squared error as the optimization criteria.

**XGBoost.** XGBoost, also known as Extreme Gradient Boosting, is an ensemble learning method that builds decision trees in a step-by-step manner, creating a strong predictive model. It minimizes errors by learning from previous model iterations and placing more emphasis on mispredicted instances. This method enables the effective capture of intricate connections within the data. XGBoost is known for its efficiency, speed, and high predictive performance, making it a valuable choice for lean body mass prediction [38].

We implemented a XGBoost regression model with a max depth of 6, learning rate of 0.3 and used a full L2 regularization term on the weights, with uniform sampling on the training instances.

**LassoNET.** Neural networks, serve as a foundational tool in machine learning, capable of discerning patterns and making predictions. LassoNet, a specific neural network framework, enhances this process by integrating feature selection and model fitting into its optimization objective. It incorporates a residual connection from input to output, allowing only features with active connections to participate in hidden layers. A lasso (L1) penalty is strategically employed in the residual layer, promoting sparsity, and thereby selecting the most relevant features [39]. This is achieved by optimizing the following objective function:

$$L(\theta, W) + \lambda||\theta||_1 \tag{5}$$

Where $L(\theta, W)$ is a mean-squared error loss function with the following constraint:

$$||W_j||_\infty \leq M|\theta_j|, j = 1, \ldots, N \qquad (6)$$

Here, $\theta$ are the weights of the residual skip layer, W are the weights of the overall network, $j$ represents the feature number, $M$ is the hierarchy parameter controlling the relative strength of the linear and nonlinear components, and $\lambda$ is the L1-penalty coefficient and is a learned parameter which controls the complexity of the fitted model.

What sets LassoNet apart is its ability to efficiently navigate a regularization path, seamlessly traversing progressive levels of sparsity through proximal gradient methods and warm restarts. This distinctive feature allows LassoNet to adaptively select varying numbers of features while simultaneously fitting flexible nonlinear neural network models. This dynamic and data-driven approach to feature selection and predictive modeling positions LassoNet as an ideal candidate for crafting lean mass prediction equations.

We implemented a LassoNET regression model with 10-fold cross-validation, a hidden layer size of 100, a hierarchy parameter (M) of 10, Adam optimizer, no dropouts, no batches, and early stoppage after 10 epochs with no improvement. Improvement was defined as at least a 1% decrease compared to the previous epoch.

**Permutation importance.**   To understand the impact of individual features on the predictive model, we analyzed the importance of features using the permutation importance method. Permutation importance is an ML technique used to evaluate the significance of features in predictive models. It involves shuffling the values of each feature independently and measuring the subsequent impact on the model's performance. The extent to which the model's accuracy decreases after permuting a specific feature reflects its importance: the greater the decrease, the more crucial the feature is for the model's predictive capacity. This method aids in identifying the most influential predictors in a model, offering insights into feature relevance, and aiding in feature selection [40]. We used ten permutations (repeats) per model for our analysis.

## Results

### Sample characteristics

Table 1 presents the summary characteristics of the sample population for the training, testing, and validation datasets. To assess differences between these datasets, we used the Yuen-Dixon tests for continuous variables and chi-square tests for categorical variables to compare the characteristics of our training/test set with our validation set. The results showed no statistically significant differences in age, weight, height, or waist circumference between the training and testing datasets ($p > 0.05$ for all comparisons). However, significant differences were observed between the NHANES (training/testing) and Look AHEAD (validation) datasets for weight ($p < 0.001$), and waist circumference ($p < 0.001$), reflecting the different population characteristics of these studies due to the inherent domain shift that exists in the data.

**Total lean body mass.**   The evaluation metrics for total lean body mass prediction are visually represented in Fig 1.

For the NHANES dataset, our models demonstrated a MAPE of [4.22, 3.14, 3.16, 3.02], and an $R^2$ of [0.73, 0.85, 0.84, 0.86] for the anthropometric equation, random forest, XGBoost and LassoNet, respectively. These results indicate almost no difference between the models, but already some improvement over the linear regression based anthropometric equation–strengthening the case for more the use of sophisticated models for this task.

For the Look AHEAD dataset, our models achieved a MAPE of [3.93, 3.76, 3.82, 3.49] and an $R^2$ of [0.67, 0.69, 0.68, 0.74] for the anthropometric equation, random forest, XGBoost and LassoNet, respectively. We see a drop in performance, as expected since the model was not

**Table 1. Sample characteristics in the NHANES and Look AHEAD datasets.**

| Categorical variables | | | | |
| --- | --- | --- | --- | --- |
| | | Total (percentage) | | |
| | | Training data | Testing data | Validation data |
| Sex | Male | 4,396 (49.7%) | 1,098 (49.6%) | 512 (37.4%) |
| | Female | 4,452 (50.3%) | 1,115 (50.4%) | 857 (62.6%) |
| Ethnicity | Hispanic or Latino | 2,223 (25.1%) | 543 (24.5%) | 376 (27.5%) |
| | Not Hispanic or Latino | 6,625 (74.9%) | 1,670 (75.5%) | 993 (72.5%) |
| | Black or African-American | 1,891 (21.4%) | 470 (21.2%) | 147 (10.7%) |
| | White | 3,972 (44.9%) | 1008 (45.5%) | 789 (57.6%) |
| | Other/mixed/not known | 762 (8.6%) | 192 (8.6%) | 57 (4.2%) |
| Continuous variables | | | | |
| Characteristic | | mean (SD) | | |
| | | Training data | Testing data | Validation data |
| Waist circum. [cm] | | 100.3 (14.9) | 100.0 (1426) | 111.0 (12.3) |
| Age [years] | | 56.5 (8.2) | 56.5 (8.2) | 58.4 (6.9) |
| Weight [kg] | | 81.8 (19.3) | 81.3 (18.9) | 96.8 (16.7) |
| Height [cm] | | 167.2 (9.9) | 167.0 (10.1) | 165.7 (9.8) |
| Lean Body Mass [kg] | | 47.8 (12.1) | 47.5 (11.7) | 53.0 (11.2) |
| Lean Body Mass Percentage [%] | | 58.4 (8.3) | 58.4 (8.2) | 54.8 (6.7) |
| Appendicular Lean Mass [kg] | | 20.8 (6.3) | 20.6 (6.1) | 22.6 (5.9) |
| Appendicular Lean Mass Percentage [%] | | 25.4 (4.7) | 25.3 (4.7) | 23.3 (5.2) |
| Appendicular Skeletal Muscle Mass [kg] | | 19.5 (6.0) | 19.4 (5.8) | 20.0 (5.3) |
| Appendicular Skeletal Muscle Mass Percentage [%] | | 23.8 (4.5) | 23.9 (4.5) | 20.6 (3.3) |

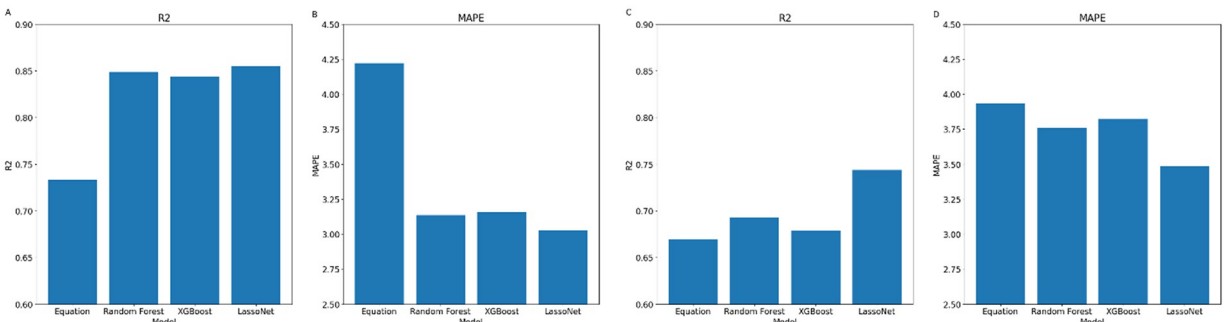

**Fig 1.** Evaluation Metrics and for Total lean body mass prediction–Evaluation of the prediction task for each of the models on the testing data (NHANES), using $R^2$ (**A**) and MAPE (**B**) metrics, as well as evaluation on the validation data (Look AHEAD) using $R^2$ (**C**) and MAPE (**D**) metrics.

trained on the Look AHEAD data, which also contains a domain shift. Additionally, while we can see the Random Forest and XGBoost models slightly outperforming the anthropometric equations, LassoNet shows a substantial jump in the prediction task, with an 11% decrease in MAPE, as well as 10% increase in $R^2$ when comparing with the equations.

In addition to the MAPE and $R^2$ value, we calculated the Root Mean Square Error (RMSE) for each model. For TLM prediction, the RMSE values were [3.38, 3.19, 3.24, 3.09] for the anthropometric equation, Random Forest, XGBoost, and LassoNet models, respectively, when applied to the NHANES testing dataset. On the Look AHEAD validation dataset, the RMSE values were [5.29, 4.59, 4.69, 4.35] for the same models.

Fig 2 shows the actual and predicted values for TLM for the three models: Random Forest, XGBoost and LassoNet, as well as the anthropometric equation to be used as reference. In each subplot, the red line represents the ground truth, which is the ideal results where the predicted values exactly match the actual values. The green line represents the fit line, which is the best linear fit to the data points.

The permutation importance plots for TLM prediction (Fig 3) show the importance of each feature in the prediction task. As expected, all models depended heavily on the subject's total weight to predict their lean mass, with sex being the second most crucial factor. Additional factors that affected the decision for all models are the waist circumference and the height.

Age was a minor factor for all the models, and unlike the other models the ethnicity also had a small effect on the prediction for the LassoNet model.

**Appendicular lean mass and appendicular skeletal muscle mass.** As part of our analysis, we predicted the ALM and ASMM to evaluate the generalizability of the methods used to predict TLM. Since the anthropometric equations were designed to predict TLM, we used linear regression as a comparison metric instead. As with the TLM prediction, all values were normalized by their corresponding body weights to provide mass percentages. The evaluation metrics for ALM and ASMM prediction are visually represented in Fig 4.

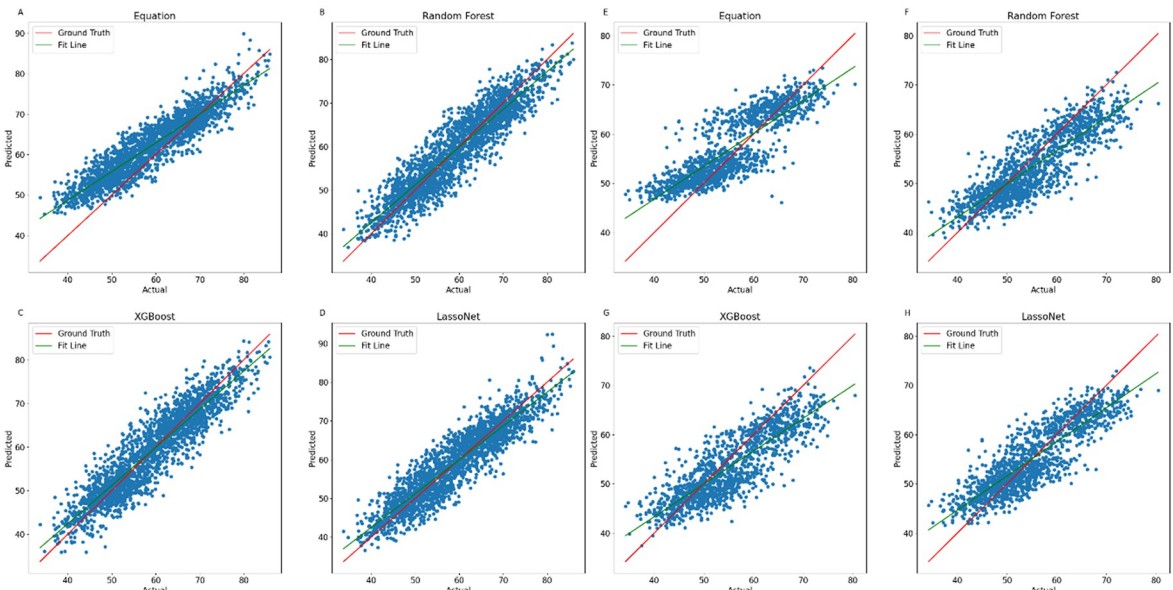

**Fig 2.** Predicted vs Actual Total Lean Body Mass–prediction of total lean body mass adjusted for weight with testing data (NHANES) for the Anthropometric equation (**A**), Random Forest (**B**), XGBoost (**C**) and LassoNet (**D**), and the validation data (Look AHEAD) for the Anthropometric equation (**E**), Random Forest (**F**), XGBoost (**G**) and LassoNet (**H**). The red line depicts a perfect linear fit, while the green line is the model's actual linear fit.

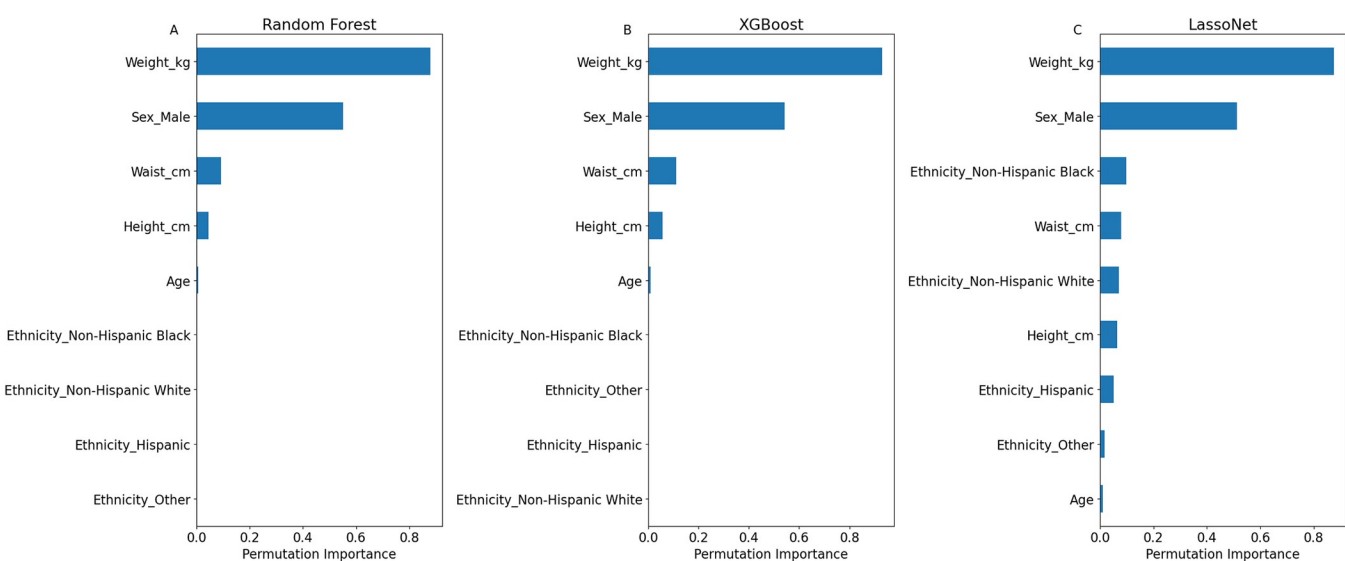

**Fig 3.** Premutation Importance plots for Total Lean Body Mass prediction–Permutation importance plots for each one of the prediction models: Random Forest (**A**), XGBoost (**B**), LassoNet (**C**).

Similarly to the TLM prediction, the results for the NHANES dataset were exceptionally good for all models, with $R^2$ values higher than 0.79 and MAPE values lower than 1.9 for all models with no substantial differences between them. However, for the Look AHEAD dataset our models demonstrated a MAPE of [1.93, 2.03, 2.04, 1.82], and an $R^2$ of [0.7, 0.67, 0.66, 0.73] for linear regression, random forest, XGBoost and LassoNet on ALM prediction, and a MAPE of [1.82, 1.88, 1.92, 1.74], and an $R^2$ of [0.68, 0.66, 0.65, 0.71] for linear regression, random forest, XGBoost and LassoNet on ASMM prediction. RMSE values were [2.45, 2.5, 2.55, 2.35] kg for the anthropometric equation, Random Forest, XGBoost, and LassoNet models, respectively, for the ALM prediction and [2.32, 2.4, 2.45, 2.24] kg for the same models in the ASMM prediction.

This demonstrates that the prediction methods do not work only for the specific TLM prediction task but can be useful for prediction of other metrics. As before, the LassoNet model performed substantially better than the other prediction methods, but in this case, we see that linear regression actually performs better than the other ML methods. This goes to show the

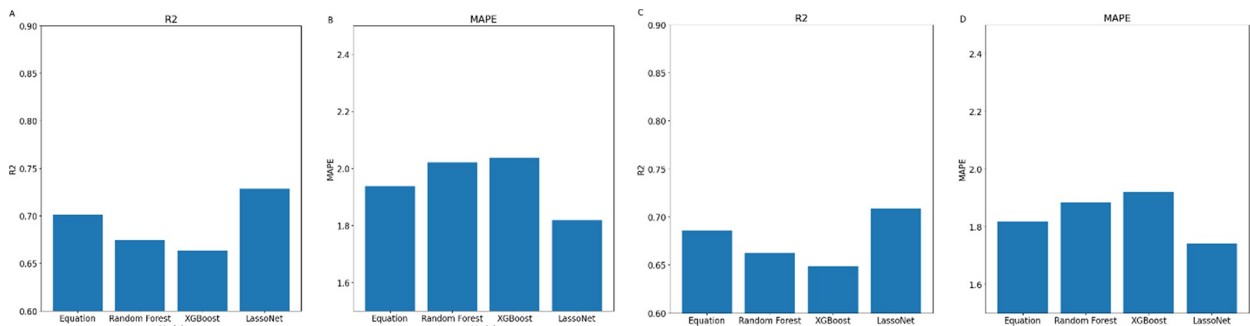

**Fig 4.** Evaluation Metrics and for Appendicular Lean Mass and Appendicular Skeletal Muscle Mass prediction–Evaluation of the prediction task for each of the models on the validation data (Look AHEAD) for prediction of Appendicular Lean Mass using $R^2$ (**A**) and MAPE (**B**) metrics, as well as evaluation on the validation data (Look AHEAD) for prediction of Appendicular Skeletal Muscle Mass using $R^2$ (**C**) and MAPE (**D**) metrics.

robustness of linear regression, as well as the benefit of using LassoNet, that outperforms all other traditional methods.

The permutation importance plots for ALM and ASMM (Fig 5) show the importance of each feature in the prediction task. Similarly to the TLM prediction, all models depended heavily on the subject's total weight to predict their lean mass, with sex being the second most crucial factor. Waist circumference played a bigger role in these predictions' tasks than in the TLM prediction, and height played a minor factor as well.

However, compared to the TLM prediction, there is more emphasis on ethnicity, with the non-Hispanic Black ethnicity being a factor for all the models. Additionally, the LassoNet model did account for other ethnicities as well, which in turn led to an increase in the model's accuracy. This was true for both the ALM and ASMM models.

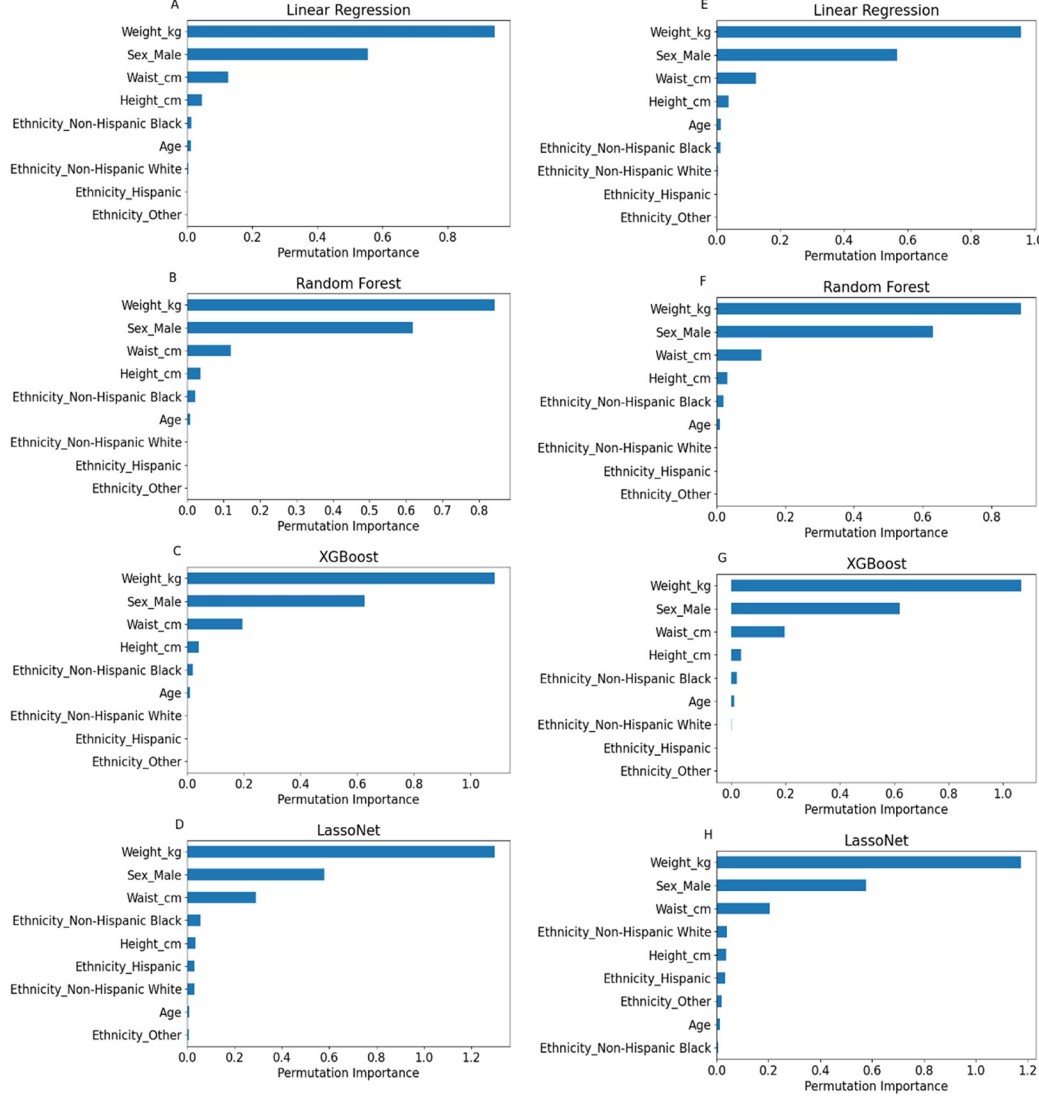

**Fig 5. Premutation Importance plots for Appendicular Lean Mass and Appendicular Skeletal Muscle Mass prediction**—Permutation importance plots for each one of the models for ALM prediction: Linear Regression (**A**) Random Forest (**B**), XGBoost (**C**), LassoNet (**D**), as well as for ASMM prediction: Linear Regression (**E**) Random Forest (**F**), XGBoost (**G**), LassoNet (**H**).

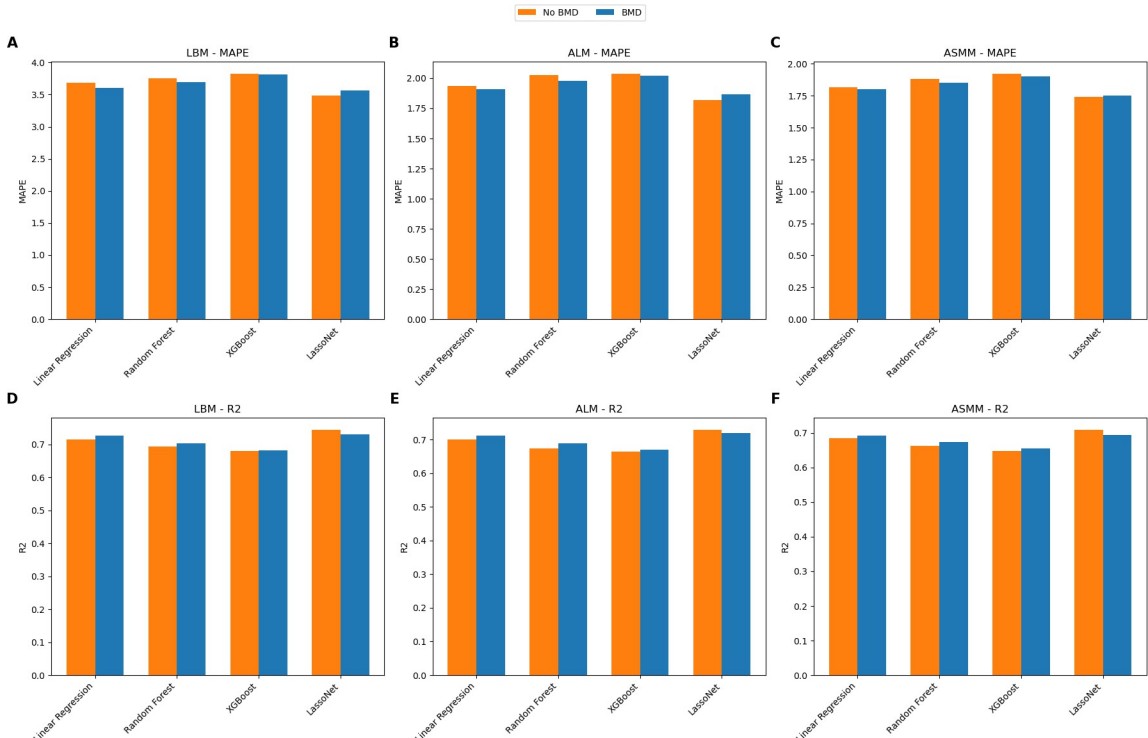

**Fig 6.** Evaluation Metrics comparison with or without Bone Mineral Density–Evaluation of the prediction task for each of the models on the validation data (Look AHEAD), with MAPE(**A-C**) and $R^2$ (**D-E**) plotted for TLM (**A, D**), ALM (**B, E**) and ASMM (**C, F**).

### Bone mineral density as a predictor

In an attempt to enhance the predictive capabilities of our models for lean mass estimation, we integrated bone mineral density (BMD) measurements along with the anthropometric data and demographic information. The hypothesis was that incorporating BMD might strengthen the model's generalizability.

Our analysis revealed that adding BMD did not enhance the models' predictive accuracy. Specifically, when assessing the performance of the LassoNet model, which emerged as the most effective model in our study, the incorporation of BMD resulted in a decrease in predictive performance for the validation dataset. As indicated in Fig 6, which evaluates the model's performance in the prediction tasks, the $R^2$ values remained consistent, showing no notable improvement with the inclusion of BMD data. Similarly, the MAPE exhibited no significant decrease, indicating that the models' predictive accuracy was not adversely affected by excluding BMD.

These outcomes suggest that while BMD is an essential metric in evaluating bone health, its incorporation into the lean mass prediction models did not enhance their predictive capacity. In fact, the inclusion of BMD data led to a deterioration in performance for the primary model and brought only marginal improvements in others, signaling that for our specific prediction task, the relevance of BMD might be limited. Therefore, we excluded BMD in our modeling.

### Discussion

The study aimed to use machine learning techniques to predict TLM, ALM and ASMM using different predictors and a myriad of methods. Our findings show that techniques like Random

Forest, XGBoost and LassoNET can predict TLM, ALM and ASMM with high accuracy. Machine learning techniques appear superior to established equations derived from statistical methods. Most importantly, BMD does not significantly influence predictions in this dataset. However, there are concerns about the generalizability of these findings to high-risk populations, as outlined below.

Methodology and algorithms have been constantly evolving and used in different clinical settings like insomnia [41]. AA Huang et al, presented a novel framework combining bootstrap simulation and SHapley Additive exPlanations (SHAP) values to enhance the interpretability of machine learning models [42].

There have been significant advancements in using machine learning for body composition assessment. Prior work includes using machine learning to predict malnutrition in older adults, involving metrics like calf circumference and a two-step approach based on the Global Initiative on Malnutrition (GLIM) criteria [43, 44]. Sarcopenia prediction has also been performed using natural language processing and electronic health records [45]. Our study is unique in applying LassoNET to a large dataset (NHANES) and cross-validating results on another large dataset (LOOK AHEAD).

NHANES data has been used to evaluate cardiovascular disease risk from heavy metal exposure [46]. The interpretable machine learning model using SHAP found significant associations between coronary heart disease (CHD) risk and exposure to heavy metals like lead and cadmium, providing actionable insights for public health interventions.

Despite the advantages of large datasets like NHANES, there is potential for machine learning models to compromise privacy by accurately identifying individuals within de-identified datasets, especially using variables like physical activity [47].

The study has several important limitations. The models lack outcome measures, which are crucial for evaluating their effectiveness in real-world clinical settings. Additionally, since the study involved analysis of existing datasets (no prospective data), it did not include cohorts highly vulnerable to muscle mass loss limiting the generalizability of the findings. The applicability of the results to cohorts that are highly vulnerable to loss of muscle mass like advanced malignancy, heart failure, COPD needs further investigation [48–50]. These limitations highlight the need for further research to enhance the applicability and effectiveness of predictive models in diverse clinical environments.

Despite these shortcomings, the data can generate robust reference ranges for the general population. With more advanced domain adaptation methods, it might be possible to apply machine learning techniques to predict muscle mass in the aforementioned vulnerable cohorts. As a technique, LassoNET appears robust in prediction accuracy.

The clinical implications of these results are multifold. First, these results shed more light on the concept of metabolically healthy obesity, suggesting that not all weight gain is detrimental [51–53]. The inadequacy of BMI for determination of individual health is further highlighted by these results [54]. Second, machine learning algorithms and networks can be adapted to incorporate individuals with advanced malignancy in different settings, evaluating muscle loss extent based on age, body weight, ethnicity, and sex reference standards [55, 56]. Third, after determination of expected muscle/lean mass, specific goals to retain and enhance muscle strength development may be pursued on a large-scale population basis. Fourth, pharmacological therapies that are based upon adiposity pharmacokinetics could be fine-tuned based upon expected muscle mass distribution, thereby limiting toxicities, and achieving adequate therapeutic results [57, 58]. This could reduce the reliance on weight-based drug dosage calculations and usher in dosage calculations based on fat and muscle percentage.

## Conclusions

The application of Machine Learning techniques for predicting lean mass displays high accuracy when considering even just anthropometric measurements, age, sex, and ethnicity-related factors. These predictive capabilities hold significant implications for chronic disease management, suggesting a promising avenue for more precise and personalized health assessments.

## Acknowledgments

The authors wish to thank the staff and participants of the Look AHEAD Study for their valuable contributions.

Look AHEAD was conducted by the Look AHEAD Research Group and supported by the National Institute of Diabetes and Digestive and Kidney Diseases (NIDDK); the National Heart, Lung, and Blood Institute (NHLBI); the National Institute of Nursing Research (NINR); the National Institute of Minority Health and Health Disparities (NIMHD); the Office of Research on Women's Health (ORWH); and the Centers for Disease Control and Prevention (CDC). The data from Look AHEAD was supplied by the NIDDK Central Repositories. This manuscript was not prepared under the auspices of the Look AHEAD and does not represent analyses or conclusions of the Look AHEAD Research Group, the NIDDK Central Repositories, or the NIH. Any opinion, findings, and conclusions or recommendations expressed in this material are those of the authors and do not necessarily reflect the views of the National Science Foundation. The data was provided to us in accordance with the NIDDK-NIH researcher data sharing agreement.

## Author Contributions

**Conceptualization:** Daniel Olshvang, Rama Chellappa.

**Data curation:** Prasanna Santhanam.

**Formal analysis:** Daniel Olshvang.

**Investigation:** Daniel Olshvang.

**Methodology:** Daniel Olshvang.

**Writing – original draft:** Daniel Olshvang.

**Writing – review & editing:** Carl Harris, Rama Chellappa, Prasanna Santhanam.

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
