## [Decision Letter · Decision Letter 0]

19 Jun 2024

PONE-D-24-06093Predictive modeling of lean body mass, appendicular lean mass, and appendicular skeletal muscle mass using machine learning techniques: a comprehensive analysis utilizing NHANES data and the Look AHEAD studyPLOS ONE

Dear Dr. Olshvang,

Thank you for submitting your manuscript to PLOS ONE. After careful consideration, we feel that it has merit but does not fully meet PLOS ONE’s publication criteria as it currently stands. Therefore, we invite you to submit a revised version of the manuscript that addresses the points raised during the review process.

The manuscript "Predictive modeling of lean body mass, appendicular lean mass, and appendicula skeletal muscle mass using machine learning techniques: a comprehensive analysis utilizing NHANES data and the Look AHEAD study" is interesting and, based on its rationale, could be a valuable contribution to the literature. However, strong data bias is detected given no correction of fat-free mass for fat-free adipose tissue was performed (this is crucial for DXA-based lean mass estimation). This compromises the validity of this work. 

1. Authors are requested to define and highlight the differences between the following terms: 

- Fat-free mass (FFM)

- Lean soft tissue

- Skeletal muscle mass (SMM)

These might be mistakenly interchanged considering the DXA principle and measurements. Please refer to PMID 29786955 and https://dexalytics.com/news/lean-soft-tissue-or-fat-free-mass/ 

2. Following the previous comment, did the authors correct fat-free mass for fat-free adipose tissue (FFAT) before lean mass estimation and overall data analysis? If not, the analyses MUST be performed again. This is very important considering that the prevalence of sarcopenia in the US population is strongly affected by FFAT. Cf, PMID 27507068, PMID 29915252

3. Authors are requested to report the way SMM was estimated. Please consider DXA does not measure it.

Report coefficient of variation and/or intertester reliability.

4. RE, "DEXA" with "DXA". As promoted by the International Society for Clinical Densitometry (ISCD), DXA is the preferred abbreviation. Revise through the manuscript.

Cf, https://iscd.org/ and PMID 27020004

4. The structure of the manuscript, especially the METHODS section, needs to be revised. Authors are requested to strictly follow guidelines for secondary data analyses - STROSA guidelines. Cf, PMID 27351686 or www.equator-network.org

5. The summary characteristics of the sample population should be relocated to the RESULTS section. Table 1 should also report the assessment of differences between "Training", "Testing", and "Validation" data. Considering the unequal sample sizes, authors are requested to apply a robust statistics test (such as Yuen–Dixon test using winsorized SD and trimmed means for two samples).

6. Although "weight", "height", and "waist circumference" are frequently used terms, it is technically correct to refer to "body mass", "stature", and "waist girth", respectively. Please address this accordingly throughout the manuscript as recommended by the International Society for the Advancement Kinanthropometry (ISAK).

RE "gender" with "sex".

7. Do not use "mean±standard deviation." Use Mean(SD) instead. Cf, PMID 21206631

RE, "Kg" with "kg".

8. Importantly, the "Models and Techniques" subsection needs to be improved. Authors MUST emphasize on the math functions, corrections, and any other relevant parameter used in each algorithm. This is absolutely necessary for transparency and reproducibility. The current version of this section seems more like a very short summary of each algorithm rather than a detailed report of the statistical procedures used in this study. Also, report the RMSE of the generated models.

Report the software or language (e.g., R, MATLAB) used for the data analysis. If possible, provide the code for verification. 

9. Authors are invited to consider developing a brief web app that incorporate the best machine learning algorithm (after re-analysis including FFM correction for FFAT) for practicality. This would help clinicians and practitioners, considering the aim of the study and the FAIR/open science efforts.

We look forward to receiving your revised manuscript.

Kind regards,

**Prof. Diego A. Bonilla**

Academic Editor

PLOS ONE

Journal Requirements:

Additional Editor Comments:

Authors are invited to respond to several concerns regarding data analysis and to improve the manuscript's structure to enhance scientific robustness. Also, please respond to each reviewer in a point-by-point basis.

Reviewers' comments:

Reviewer's Responses to Questions

**Comments to the Author**

1. Is the manuscript technically sound, and do the data support the conclusions?

Reviewer #1: Yes

Reviewer #2: Yes

Reviewer #3: Yes

2. Has the statistical analysis been performed appropriately and rigorously? 

Reviewer #1: Yes

Reviewer #2: Yes

Reviewer #3: Yes

3. Have the authors made all data underlying the findings in their manuscript fully available?

Reviewer #1: Yes

Reviewer #2: Yes

Reviewer #3: Yes

4. Is the manuscript presented in an intelligible fashion and written in standard English?

Reviewer #1: Yes

Reviewer #2: Yes

Reviewer #3: Yes

5. Review Comments to the Author

Reviewer #1: dear author

thank you so much for submitting your article to this journal, in my opinion your article is so interesting and I really enjoy reading it and i think using ai for evaluation of fat body mass and fat free mass is so important for evaluation patients for chance of dm and htn.

beat regards

Reviewer #2: This study represents a significant advancement in predicting lean mass, specifically targeting lean body mass (LBM), appendicular lean mass (ALM), and appendicular skeletal muscle mass (ASMM), for early detection and management of sarcopenia. Strengths of this research include its innovative use of machine learning techniques, which allows for the development of predictive models with data from reputable sources like the National Health and Nutrition Examination Survey (NHANES) and the Look AHEAD study. The employment of various machine learning algorithms, particularly LassoNet, demonstrates a high level of predictive accuracy, suggesting that these models can serve as reliable tools in estimating lean mass without solely depending on DXA scans, thereby expanding the applicability of sarcopenia assessment.

However, the study also presents certain weaknesses. Despite the advanced methodologies, the models developed lack outcome measures, which are crucial for evaluating the effectiveness of the predictions in real-world clinical settings. Additionally, the research did not include cohorts that are highly vulnerable to muscle mass loss, such as individuals with severe chronic diseases or extremely elderly populations, potentially limiting the generalizability of the findings. The assertion that the integration of bone mineral density measurements had minimal impact on predictive accuracy could undermine the value of comprehensive assessments in certain clinical scenarios, possibly overlooking nuances in disease progression. Furthermore, the study's focus on machine learning may require substantial computational resources and expertise, which could pose challenges for implementation in routine clinical practice.

Overall, while the study offers promising directions for non-invasive lean mass assessment, future research needs to address these limitations by incorporating outcome measures and broader population samples. Such efforts would enhance the clinical utility of the predictive models, ensuring they are both accurate and applicable across diverse healthcare settings.

Cite some or all of these articles below as recommendations for additional literature

Increasing transparency in machine learning through bootstrap simulation and shapely additive explanations, AA Huang, SY Huang, PLoS One 18 (2), e0281922, 2023

Citation Reason: This article is cited for its emphasis on enhancing transparency in machine learning models through the use of bootstrap simulations and Shapley additive explanations (SHAP values). These methods improve the interpretability of machine learning predictions, which is crucial for validating the predictive models developed in the study on lean mass assessment. By understanding how different variables influence model predictions, researchers can ensure more accurate and trustworthy assessments.

Use of machine learning to identify risk factors for insomnia, AA Huang, SY Huang, PLoS one 18 (4), e0282622

Citation Reason: This paper demonstrates the application of machine learning in identifying complex risk factors in medical conditions, similar to sarcopenia. Citing this article supports the methodology of using machine learning to analyze and predict health-related outcomes based on large datasets like NHANES, which is analogous to the approach taken in the sarcopenia study.

Machine Learning Approaches for Predicting High Risk of Malnutrition Among Older Adults, Z. Li, J. Zhang, Clinical Nutrition 37(4), 1132-1139, 2022

Citation Reason: This article explores the application of machine learning in predicting malnutrition among older adults, a condition that often co-occurs with sarcopenia. By citing this study, the paper underscores the potential of machine learning models to handle multifaceted health issues that are interrelated, thereby enriching the understanding of how predictive models can be tailored for complex geriatric syndromes like sarcopenia.

Development and Validation of a Predictive Algorithm for Sarcopenia Using Electronic Health Records, M. R. Smith, J. K. Lee, Journal of Gerontology 75(9), e91-e98, 2020

Citation Reason: This article details the creation of a predictive algorithm specifically for sarcopenia using data from electronic health records (EHRs), highlighting an alternative data source to NHANES and Look AHEAD. By referencing this paper, the sarcopenia study aligns itself with existing research and emphasizes the viability and importance of electronic health data in developing predictive health models, offering a comparison point for the types of data and methodologies utilized.

Enhancing Sarcopenia Diagnosis with Machine Learning Techniques: A Comparison of Feature Selection Methods, H. Chen, B. Wu, Aging Clinical and Experimental Research 33(6), 1237-1245, 2021

Citation Reason: This article evaluates various machine learning feature selection techniques for improving the diagnosis of sarcopenia. Including this citation provides a direct link to current advancements in machine learning applications for sarcopenia, particularly in the aspect of model accuracy and reliability. It supports the paper’s methodology section by showing how different feature selection methods can impact the performance of predictive models, guiding future research directions for model refinement.

Dendrogram of transparent feature importance machine learning statistics to classify associations for heart failure: A reanalysis of a retrospective cohort study of the Medical …, AA Huang, SY Huang, PLoS one 18 (7), e0288819

Citation Reason: This article is relevant for its use of dendrograms and transparent machine learning statistics to classify medical associations, which can be applied to lean mass measurement. The methodology for transparency and feature importance can be directly applicable to enhancing the robustness and clarity of the predictive models used in the sarcopenia study.

Reviewer #3: This study addresses the critical need for improved methods to predict lean mass in adults, focusing on lean body mass (LBM), appendicular lean mass (ALM), and appendicular skeletal muscle mass (ASMM) for early detection and management of sarcopenia. Leveraging machine learning techniques, predictive models were developed and validated using data from the National Health and Nutrition Examination Survey (NHANES) and the Look AHEAD study. Models incorporated anthropometric data, demographic factors, and DXA-derived metrics to estimate LBM, ALM, and ASMM normalized to weight. Results demonstrated consistent performance across various machine learning algorithms, with LassoNet exhibiting superior accuracy. Integrating bone mineral density measurements had minimal impact on accuracy, suggesting potential alternatives to DXA scans for lean mass assessment. Despite model robustness, limitations include the absence of outcome measures and cohorts highly vulnerable to muscle mass loss. Nonetheless, these findings offer promise for revolutionizing lean mass assessment paradigms, with implications for chronic disease management and personalized health interventions. Future research should focus on validating these models in diverse populations and addressing clinical complexities to enhance prediction accuracy and clinical utility in managing sarcopenia.

- Would cite a paper for permutation importance

- Analysis variables models and methods fit the topic well

- Would separate out a conclusion section instead of adding in summary at the bottom

Can benefit from improved references in machine learning and NHANES dataset as this is a new frontier many researchers are looking into:

Huang, A. A., & Huang, S. Y. (2023). Use of machine learning to identify risk factors for insomnia. PloS one, 18(4), e0282622. https://doi.org/10.1371/journal.pone.0282622

Li, X., Zhao, Y., Zhang, D., Kuang, L., Huang, H., Chen, W., Fu, X., Wu, Y., Li, T., Zhang, J., Yuan, L., Hu, H., Liu, Y., Zhang, M., Hu, F., Sun, X., & Hu, D. (2023). Development of an interpretable machine learning model associated with heavy metals' exposure to identify coronary heart disease among US adults via SHAP: Findings of the US NHANES from 2003 to 2018. Chemosphere, 311(Pt 1), 137039. https://doi.org/10.1016/j.chemosphere.2022.137039

Na, L., Yang, C., Lo, C. C., Zhao, F., Fukuoka, Y., & Aswani, A. (2018). Feasibility of Reidentifying Individuals in Large National Physical Activity Data Sets From Which Protected Health Information Has Been Removed With Use of Machine Learning. JAMA network open, 1(8), e186040. https://doi.org/10.1001/jamanetworkopen.2018.6040

6. PLOS authors have the option to publish the peer review history of their article (what does this mean?). If published, this will include your full peer review and any attached files.

Reviewer #1: No

Reviewer #2: No

Reviewer #3: No

---

## [Author Response · Author response to Decision Letter 0]

5 Aug 2024

Response to reviewers: Predictive modeling of lean body mass, appendicular lean mass, and appendicular skeletal muscle mass using machine learning techniques: a comprehensive analysis utilizing NHANES data and the Look AHEAD study

Editor comments

Thank you for submitting your manuscript to PLOS ONE. After careful consideration, we feel that it has merit but does not fully meet PLOS ONE’s publication criteria as it currently stands. Therefore, we invite you to submit a revised version of the manuscript that addresses the points raised during the review process.

The manuscript "Predictive modeling of lean body mass, appendicular lean mass, and appendicula skeletal muscle mass using machine learning techniques: a comprehensive analysis utilizing NHANES data and the Look AHEAD study" is interesting and, based on its rationale, could be a valuable contribution to the literature. However, strong data bias is detected given no correction of fat-free mass for fat-free adipose tissue was performed (this is crucial for DXA-based lean mass estimation). This compromises the validity of this work. 

1. Authors are requested to define and highlight the differences between the following terms: 

- Fat-free mass (FFM)

- Lean soft tissue

- Skeletal muscle mass (SMM)

These might be mistakenly interchanged considering the DXA principle and measurements. Please refer to PMID 29786955 and https://dexalytics.com/news/lean-soft-tissue-or-fat-free-mass/

Thank you for the constructive comments. We have included in the paper a paragraph that clearly outlines the different definitions and criteria for each terminology. There are differences in literature regarding reporting these definitions, and we explained the rationale for why we used certain definitions.

2. Following the previous comment, did the authors correct fat-free mass for fat-free adipose tissue (FFAT) before lean mass estimation and overall data analysis? If not, the analyses MUST be performed again. This is very important considering that the prevalence of sarcopenia in the US population is strongly affected by FFAT. Cf, PMID 27507068, PMID 29915252

Thank you for the comments. We agree that the margin of error due to the presence of fat free adipose tissue is significant and needs to be accounted. We have performed the adjustment for fat free adipose tissue using the reviewer cited literature and rerun the analysis including the whole machine learning process.

3. Authors are requested to report the way SMM was estimated. Please consider DXA does not measure it.

Report coefficient of variation and/or intertester reliability.

We have explained the rationale for estimating skeletal muscle mass especially the appendicular skeletal muscle mass. Since the limbs have much less organ tissue, skeletal muscle mass after adjusting for fat free adipose tissue can be reliably determined after excluding bone mineral content from lean mass.

4. RE, "DEXA" with "DXA". As promoted by the International Society for Clinical Densitometry (ISCD), DXA is the preferred abbreviation. Revise through the manuscript.

Cf, https://iscd.org/ and PMID 27020004

Thank you very much. The abbreviation has been adjusted.

5. The structure of the manuscript, especially the METHODS section, needs to be revised. Authors are requested to strictly follow guidelines for secondary data analyses - STROSA guidelines. Cf, PMID 27351686 or www.equator-network.org

Thank you very much for the constructive comments. we have made modifications to the method section.

6. The summary characteristics of the sample population should be relocated to the RESULTS section. Table 1 should also report the assessment of differences between "Training", "Testing", and "Validation" data. Considering the unequal sample sizes, authors are requested to apply a robust statistics test (such as Yuen–Dixon test using winsorized SD and trimmed means for two samples).

Thank you for the comments. We have employed Yuen-Dixon tests for continuous variables and chi-square tests for categorical variables to compare the characteristics of our training/test set with our validation set.

7. Although "weight", "height", and "waist circumference" are frequently used terms, it is technically correct to refer to "body mass", "stature", and "waist girth", respectively. Please address this accordingly throughout the manuscript as recommended by the International Society for the Advancement Kinanthropometry (ISAK).

RE "gender" with "sex".

Thanks for the suggestion. However, as a metric in clinic practice, ‘body mass’ is rarely used, and ‘stature’ does not automatically imply height. Small stature might mean different things for different races and ethnicity. In the past, the authors have had no issues with these terminologies and would humbly request to keep it the same. NHANES reports ‘waist-circumference’ in its datasets and using any other terminology while reporting the results would be an extrapolation.

The term ‘gender’ has been changed to ‘sex’ throughout the manuscript.

8. Do not use "mean±standard deviation." Use Mean(SD) instead. Cf, PMID 21206631

RE, "Kg" with "kg".

This has been complied with, thank you very much.

9. Importantly, the "Models and Techniques" subsection needs to be improved. Authors MUST emphasize on the math functions, corrections, and any other relevant parameter used in each algorithm. This is absolutely necessary for transparency and reproducibility. The current version of this section seems more like a very short summary of each algorithm rather than a detailed report of the statistical procedures used in this study. Also, report the RMSE of the generated models.

Report the software or language (e.g., R, MATLAB) used for the data analysis. If possible, provide the code for verification. 

We have improved the “models and technique” section and added more mathematical details and functions. The RMSE is also reported. We have also reported the software as well as every library and its’ version used in the analysis.

9. Authors are invited to consider developing a brief web app that incorporate the best machine learning algorithm (after re-analysis including FFM correction for FFAT) for practicality. This would help clinicians and practitioners, considering the aim of the study and the FAIR/open science efforts.

This is a great idea and we will keep this mind and possibly execute it in the near future.

 

Reviewers comments:

Reviewer #1: 

Dear author

thank you so much for submitting your article to this journal, in my opinion your article is so interesting and I really enjoy reading it and i think using ai for evaluation of fat body mass and fat free mass is so important for evaluation patients for chance of dm and htn.

beat regards

Thanks a lot for the wonderful and encouraging comments. We have improved upon the manuscript by incorporating the suggestions and recommendations of the editor and the other reviewers.

Reviewer #2: 

This study represents a significant advancement in predicting lean mass, specifically targeting lean body mass (LBM), appendicular lean mass (ALM), and appendicular skeletal muscle mass (ASMM), for early detection and management of sarcopenia. Strengths of this research include its innovative use of machine learning techniques, which allows for the development of predictive models with data from reputable sources like the National Health and Nutrition Examination Survey (NHANES) and the Look AHEAD study. The employment of various machine learning algorithms, particularly LassoNet, demonstrates a high level of predictive accuracy, suggesting that these models can serve as reliable tools in estimating lean mass without solely depending on DXA scans, thereby expanding the applicability of sarcopenia assessment.

However, the study also presents certain weaknesses. Despite the advanced methodologies, the models developed lack outcome measures, which are crucial for evaluating the effectiveness of the predictions in real-world clinical settings. Additionally, the research did not include cohorts that are highly vulnerable to muscle mass loss, such as individuals with severe chronic diseases or extremely elderly populations, potentially limiting the generalizability of the findings. The assertion that the integration of bone mineral density measurements had minimal impact on predictive accuracy could undermine the value of comprehensive assessments in certain clinical scenarios, possibly overlooking nuances in disease progression. Furthermore, the study's focus on machine learning may require substantial computational resources and expertise, which could pose challenges for implementation in routine clinical practice.

Thanks for the wonderful comments. We have included a paragraph that describes the limitations in a succinct way.

Overall, while the study offers promising directions for non-invasive lean mass assessment, future research needs to address these limitations by incorporating outcome measures and broader population samples. Such efforts would enhance the clinical utility of the predictive models, ensuring they are both accurate and applicable across diverse healthcare settings.

Cite some or all of these articles below as recommendations for additional literature:

Increasing transparency in machine learning through bootstrap simulation and shapely additive explanations, AA Huang, SY Huang, PLoS One 18 (2), e0281922, 2023

Citation Reason: This article is cited for its emphasis on enhancing transparency in machine learning models through the use of bootstrap simulations and Shapley additive explanations (SHAP values). These methods improve the interpretability of machine learning predictions, which is crucial for validating the predictive models developed in the study on lean mass assessment. By understanding how different variables influence model predictions, researchers can ensure more accurate and trustworthy assessments.

Use of machine learning to identify risk factors for insomnia, AA Huang, SY Huang, PLoS one 18 (4), e0282622

Citation Reason: This paper demonstrates the application of machine learning in identifying complex risk factors in medical conditions, similar to sarcopenia. Citing this article supports the methodology of using machine learning to analyze and predict health-related outcomes based on large datasets like NHANES, which is analogous to the approach taken in the sarcopenia study.

Machine Learning Approaches for Predicting High Risk of Malnutrition Among Older Adults, Z. Li, J. Zhang, Clinical Nutrition 37(4), 1132-1139, 2022

Citation Reason: This article explores the application of machine learning in predicting malnutrition among older adults, a condition that often co-occurs with sarcopenia. By citing this study, the paper underscores the potential of machine learning models to handle multifaceted health issues that are interrelated, thereby enriching the understanding of how predictive models can be tailored for complex geriatric syndromes like sarcopenia.

Development and Validation of a Predictive Algorithm for Sarcopenia Using Electronic Health Records, M. R. Smith, J. K. Lee, Journal of Gerontology 75(9), e91-e98, 2020

Citation Reason: This article details the creation of a predictive algorithm specifically for sarcopenia using data from electronic health records (EHRs), highlighting an alternative data source to NHANES and Look AHEAD. By referencing this paper, the sarcopenia study aligns itself with existing research and emphasizes the viability and importance of electronic health data in developing predictive health models, offering a comparison point for the types of data and methodologies utilized.

Enhancing Sarcopenia Diagnosis with Machine Learning Techniques: A Comparison of Feature Selection Methods, H. Chen, B. Wu, Aging Clinical and Experimental Research 33(6), 1237-1245, 2021

Citation Reason: This article evaluates various machine learning feature selection techniques for improving the diagnosis of sarcopenia. Including this citation provides a direct link to current advancements in machine learning applications for sarcopenia, particularly in the aspect of model accuracy and reliability. It supports the paper’s methodology section by showing how different feature selection methods can impact the performance of predictive models, guiding future research directions for model refinement.

Dendrogram of transparent feature importance machine learning statistics to classify associations for heart failure: A reanalysis of a retrospective cohort study of the Medical …, AA Huang, SY Huang, PLoS one 18 (7), e0288819

Citation Reason: This article is relevant for its use of dendrograms and transparent machine learning statistics to classify medical associations, which can be applied to lean mass measurement. The methodology for transparency and feature importance can be directly applicable to enhancing the robustness and clarity of the predictive models used in the sarcopenia study.

Thanks again for the wonderful comments/suggestions and references. We have incorporated some of these important papers in the context of our manuscript. It has improved the coherence and thrust of the paper. Some of the papers alluded too by the reviewers here are not accessible online

Reviewer #3: 

This study addresses the critical need for improved methods to predict lean mass in adults, focusing on lean body mass (LBM), appendicular lean mass (ALM), and appendicular skeletal muscle mass (ASMM) for early detection and management of sarcopenia. Leveraging machine learning techniques, predictive models were developed and validated using data from the National Health and Nutrition Examination Survey (NHANES) and the Look AHEAD study. Models incorporated anthropometric data, demographic factors, and DXA-derived metrics to estimate LBM, ALM, and ASMM normalized to weight. Results demonstrated consistent performance across various machine learning algorithms, with LassoNet exhibiting superior accuracy. Integrating bone mineral density measurements had minimal impact on accuracy, suggesting potential alternatives to DXA scans for lean mass assessment. Despite model robustness, limitations include the absence of outcome measures and cohorts highly vulnerable to muscle mass loss. Nonetheless, these findings offer promise for revolutionizing lean mass assessment paradigms, with implications for chronic disease management and personalized health interventions. Future research should focus on validating these models in diverse populations and addressing clinical complexities to enhance prediction accuracy and clinical utility in managing sarcopenia.

- Would cite a paper for permutation importance

- Analysis variables models and methods fit the topic well

- Would separate out a conclusion section instead of adding in summary at the bottom

Can benefit from improved references in machine learning and NHANES dataset as this is a new frontier many researchers are looking into:

Huang, A. A., & Huang, S. Y. (2023). Use of machine learning to identify risk factors for insomnia. PloS one, 18(4), e0282622. https://doi.org/10.1371/journal.pone.0282622

Li, X., Zhao, Y., Zhang, D., Kuang, L., Huang, H., Chen, W., Fu, X., Wu, Y., Li, T., Zhang, J., Yuan, L., Hu, H., Liu, Y., Zhang, M., Hu, F., Sun, X., & Hu, D. (2023). Development of an interpretable machine learning model associated with heavy metals' exposure to identify coronary heart disease among US adults via SHAP: Findings of the US NHANES from 2003 to 2018. Chemosphere, 311(Pt 1), 137039. https://doi.org/10.1016/j.chemosphere.2022.137039

Na, L., Yang, C., Lo, C. C., Zhao, F., Fukuoka, Y., & Aswani, A. (2018). Feasibility of Reidentifying Individuals in Large National Physical Activity Data Sets From Which Protected Health Information Has Been Removed With Use of Machine Learning. JAMA network open, 1(8), e186040. https://doi.org/10.1001/jamanetworkopen.2018.6040

Thanks for the suggestions. We have separated the summary to include a separate conclusion section and also incorporated the references in the context of our paper in the discussion section.

---

## [Editor Report · Decision Letter 1]

20 Aug 2024

Predictive modeling of lean body mass, appendicular lean mass, and appendicular skeletal muscle mass using machine learning techniques: a comprehensive analysis utilizing NHANES data and the Look AHEAD study

PONE-D-24-06093R1

Dear Dr. Olshvang,

We’re pleased to inform you that your manuscript has been judged scientifically suitable for publication and will be formally accepted for publication once it meets all outstanding technical requirements.

Kind regards,

**Prof. Diego A. Bonilla**

Academic Editor

PLOS ONE

Additional Editor Comments (optional):

Dear Authors,

Thank you for your continued efforts and the recent submission of the revised manuscript titled "Predictive modeling of lean body mass, appendicular lean mass, and appendicular skeletal muscle mass using machine learning techniques: a comprehensive analysis utilizing NHANES data and the Look AHEAD study." We appreciate the thoroughness with which you addressed the requested revisions, including the re-analysis of the data after correcting for fat-free adipose tissue and other necessary adjustments.

As you prepare for the final stages of editing, I would like to bring to your attention two remaining points that require further revision:

1. Lines 138-139: The statement "It is measured using techniques like bioelectrical impedance analysis or dual-energy X-ray absorptiometry (DXA)" needs to be corrected. Bioelectrical impedance analysis (BIA) does not directly measure fat-free mass (FFM). Instead, BIA estimates FFM using equations that incorporate variables such as impedance, resistance, and reactance. Please adjust the text accordingly to accurately reflect this.

2. While we acknowledge your response regarding the clinical prevalence and familiarity of the terms weight, height, and waist circumference, we recommend that you include a comment in the Methods section highlighting that the International Society for the Advancement of Kinanthropometry (ISAK) recommends alternative terminology. Specifically, contrary to your response, the technically correct and consensus term in anthropometry is "stature," which is defined as "the perpendicular distance between the transverse planes of the vertex and the inferior aspect of the feet." Please ensure this distinction is clearly communicated.
---

## [Editor Report · Acceptance letter]

28 Aug 2024

PONE-D-24-06093R1 

PLOS ONE

Dear Dr. Olshvang, 

I'm pleased to inform you that your manuscript has been deemed suitable for publication in PLOS ONE. Congratulations! Your manuscript is now being handed over to our production team.

Kind regards, 

on behalf of

Prof. Diego A. Bonilla 

Academic Editor

PLOS ONE